# QRGAN: QUANTILE REGRESSION GENERATIVE ADVERSARIAL NETWORKS

## ABSTRACT

Learning high-dimensional probability distributions by competitively training generative and discriminative neural networks is a prominent approach of Generative Adversarial Networks (GANs) among generative models to model complex real-world data. Nevertheless, training GANs likely suffer from non-convergence problem, mode collapse and gradient explosion or vanishing. Least Squares GAN (LSGANs) and Wasserstein GANs (WGAN) are of representative variants of GANs in literature that diminish the inherent problems of GANs by proposing the modification methodology of loss functions. However, LSGANs often fall into local minima and cause mode collapse. While WGANs unexpectedly encounter with inefficient computation and slow training due to its constraints in Wasserstein distance approximation. In this paper, we propose Quantile Regression GAN (QR-GAN) in which quantile regression is adopted to minimize 1-Wasserstein distance between real and generated data distribution as a novel approach in modification of loss functions for improvement of GANs. To study the culprits of mode collapse problem, the output space of discriminator and gradients of fake samples are analyzed to see if the discriminator guides the generator well. And we found that the discriminator should not be bounded to specific numbers. Our proposed QRGAN exposes high robustness against mode collapse problem. Furthermore, QRGAN obtains an apparent improvement in the evaluation and comparison of Frechet Inception Distance (FID) for generation performance assessment compared to existing variants of GANs.

## 1 INTRODUCTION

Deep learning-based data generation techniques have proved their successes in many real-world applications. Thanks to the rising of generative models, the generation of audio, images and videos, either unconditionally or conditionally, has achieved remarkable advancements in recent years. Text or structured data can be generated easily as well in many recent studies. Data generation techniques bring about efficency and creativity in human activities at every conner of the world.

Among the most influent and successful methods for data generation experiments, Variational Autoencoders (VAE) Kingma & Welling (2014) and Generative Adversarial Networks (GANs) Goodfellow et al. (2014) are the fundamental representatives of generative models.

**Variational Autoencoders (VAE):** VAEs regularize the encoder output to be a known distribution. This regularization is applied to each sample. For latent variable $\mathbf{z}$ and input $\mathbf{x}$, $p(z|x)$, not $p(z)$, is pushed to the prior distribution. With the additional reconstruction loss, the two objectives may conflict each other. Usually, mean square error (MSE) loss is used for the reconstruction loss. Because if the global minimum of MSE loss is at the expected value of the distribution, the decoders generate blurry outputs. PixelVAE Gulrajani et al. (2016) fixes VAE's blurry output by replacing MSE loss by PixelCNN van den Oord et al. (2016) decoder.

**Generative Adversarial Networks (GANs):** GANs regularize the entire decoder input distribution to be a known distribution. In other words, GANs regularize $p(z)$, not $p(z|x)$. Therefore, generators can generate sharp outputs without the conflicting objectives in VAEs. A number of GAN variants has been presented in literature to improve the data generation including Least-square

GANs (LSGANs) Mao et al. (2019), Wasserstein GANs (WGANs) Arjovsky et al. (2017), Latent-GAN Prykhodko et al. (2019), Adversarial Autoencoders (AAEs) Makhzani et al. (2016). LSGANs and WGANs improved the objective function for better generation quality. LatentGAN generates latent variables for the given autoencoder using GAN methods. AAEs replace the KL regularization from VAEs by a discriminator that distinguishes the encoder output distribution (generated data) and known distribution (real data). Although GANs can obtain better generation output, they are difficult to train because of the framework architecture in which two networks compete. Mode collapse is a common failure in GAN frameworks that generator outputs very similar samples only. Mode collapse is caused by unstable training and improper loss function.

**Variants of GANs:** Recent work targets to find the best method to train GANs with better quality and less mode collapse. The work Wiatrak et al. (2020) shows a huge number of existing studies that put an effort to improve GANs. The work also explains GANs' recent successes and problems. Accordingly, several GAN variants are introduced to stabilize the GAN training such as DCGAN Radford et al. (2016), LSGAN, WGAN-GP Gulrajani et al. (2017), and Fisher GAN Mroueh & Sercu (2017). In Least Square Generative Adversarial Networks Mao et al. (2017); Mao et al. (2019) (LSGANs), the authors propose to use mean square error (MSE) which does not saturate. Also, they found that training generator to make samples near the decision boundary instead of trying to overwhelm the discriminator results better models. Alternatively, Wasserstein GANs Arjovsky et al. (2017) applies Lipschitz condition to approximate 1-Wasserstein distance. In the original WGAN, they implement the constraint by clipping weights of critic network into a range specified by hyperparameters. WGAN-GP is introduced to improve training of WGAN. WGAN-GP give gradient penalty to push weight gradient norm of critic to 1. Thus, the generator can easily escape from local minima. However, it takes more time for computation of gradients for random inputs.

**Reinforcement Learning (RL):** is to learn the optimal policy in a given environment. The agent read an observation (state) from the environment, calculate the optimal action, and take the action. The environment returns the next state with the reward by the action. By trial and error, the agent learns the optimal action for each state and gradually reach to the optimal policy. For given time $t$, state $\mathbf{s}$, action $\mathbf{a}$, decay factor $\gamma$, and reward at $t$ is $R_t$, $Q - value$ is defined as $Q(s, a) = \mathbb{E}[\sum_{t=0}^{\infty} \gamma^t R_t | s, a]$, which in the other words is the expected value of cumulative sum of decaying rewards. Previously, Q-Learning used a table to store every Q-value for each state. This method cannot be used for complex environments which have a great deal of states.

**Quantile Regression Deep Q Network (QR-DQN:)** To learn a generalized state-value function, Deep Q Network (DQN) successfully applies deep neural network to predict Q-value Mnih et al. (2015). With some tricks for stability, DQN succeeds to beat human level performance in over 29 of 49 Atari 2600 games. Meanwhile, there are attempts to learn a distribution of Q-value: C51 Bellemare et al. (2017), Quantile Regression-Deep Q Network (QR-DQN) Dabney et al. (2017). C51 introduces the importance of distributional reinforcement learning (distributional RL) which learns Q-value distribution instead of the expected value. Distributional RL improves training stability and performance. However, the KL divergence used in C51 is not mathematically guaranteed to converge. In economics, the mean predicting quantile values is not only important. The work Koenker (2005) proposes a method called quantile regression to predict quantile values. QR-DQN demonstrates a solution that guarantees mathematical convergence by quantile regression that minimizes 1-Wasserstein distance without bias.

**Our contributions:** Adropting the above-mentioned quantile regresstion approach in this work, we propose QRGAN, a GAN-based generative model adopting quantile regresstion, to minimize 1-Wasserstein distance between real samples and generated samples using quantile regression. We train the discriminator to predict quantile values of realisticity using quantile regression. Then, we train our generator to minimize the difference of quantile values of real and fake samples to minimize 1-Wasserstein distance between the two. We analyze and compare the discriminator output space for each method to find out if discriminator can guide the generator well. Discriminators whose target is specified tend not to make ambiguous outputs. Those discriminators create sharp minima, so generators may not learn from them. For example, when a generator generates fake samples near a real sample $A$, a discriminator lowers the probability of samples near $A$ (which can be seen as "realisticity"). Then, generator should generate fake samples near another real sample $B$. However,

discriminator detects fake samples between $A$ and $B$ almost completely. Although the generator should move to $B$, the generator chooses to stay near $A$ because the way from $A$ to $B$ is too risky (the realisticity can decrease). In other words, bounded discriminator creates sharp local minima, which prevents the generator from learning. This also stops the networks from escaping from the mode collapse. Not to let the discriminator get bounded, we desire to minimize 1-Wasserstein distance. Wasserstein GAN Arjovsky et al. (2017) applies Lipschitz condition to a discriminator to approximate 1-Wasserstein distance causing a slowness in training. Our approach of quantile regression can obtain relatively faster training speed in comparision to existing variants of GANs.

The paper is organized as follows. In Section 1, we brief current state of the art and motivation of this work. In Section 2, we comprehensively present the development of QRGAN. And experimental demonstrations are shown in Section 3 including experiments on the mixture of gussian dataset and image generation experiments. Lastly, the paper is concluded in Section 4.

## 2    QUANTILE REGRESSION GAN

GANs train generative model by minimizing Kullback-Libeler (KL) divergence between real samples and generated samples. However, KL divergence is not mathematically guaranteed to converge. Thus, the discriminator often creates sharp local minima and even cause mode collapse issue. Although WGANs instead minimize 1-Wasserstein distance to fix this problem, the training is slow because Lipschitz condition is added to the discriminator (critic) network for Wasserstein distance approximation. We propose QRGAN, a GAN-based generative method, to minimize 1-Wasserstein distance between real samples and generated samples using quantile regression. We train our discriminator to predict quantile values of realisticity using quantile regression. Then, we train our generator to minimize the difference of quantile values of real and fake samples to minimize 1-Wasserstein distance between the two. First, we explain the relation of quantile regression and 1-Wasserstein distance in Section 2.1. In Section 2.2, we explain how to learn quantile values by applying quantile regression to the discriminator, and how the generator minimizes 1-Wasserstein distance. Finally, we compare QRGAN with other state of the art GAN methods in Section 2.3.

### 2.1    QUANTILE REGRESSION FOR MINIMIZATION OF 1-WASSERSTEIN DISTANCE

For given quantile fraction $\tau$ inverse CDF function $F$, distributions $\mathbb{P}_r$ and $\mathbb{P}_g$, $p - Wasserstein$ distance is defined by,

$$W_p(\mathbb{P}_r, \mathbb{P}_g) = \left( \int_0^1 \left| F_{\mathbb{P}_r}^{-1}(\tau) - F_{\mathbb{P}_g}^{-1}(\tau) \right|^p d\tau \right)^{\frac{1}{p}} \tag{1}$$

Then 1-Wasserstein distance is,

$$W_p(\mathbb{P}_r, \mathbb{P}_g) = \int_0^1 \left| F_{\mathbb{P}_r}^{-1}(\tau) - F_{\mathbb{P}_g}^{-1}(\tau) \right| d\tau \tag{2}$$

. Here, minimizing 1-Wasserstein distance is same to minimizing distance between quantile values. This method is used to indirectly minimize 1-Wasserstein distance in QR-DQN. Quantile regression is done by training the network using quantile loss. It is crafted function to have minimum value at the given quantile fraction $\tau$. Given quantile fraction $\tau \in [0, 1]$ and error $u$, quantile loss is formulated as follows,

$$\rho_\tau(u) = \begin{cases} u.(\tau - 1), & u \leq 0 \\ u.\tau, & u > 0 \end{cases} \tag{3}$$

### 2.2    QRGAN

We define the number of quantile values $N$, then quantile fractions are $\tau_0 = 0, \tau_i = \frac{i}{N+1}, \tau_N = 1$ where, $i = 1, .., N$. $1 - Wasserstein$ distance can be represented better by using middle points $\hat{\tau}$ Dabney et al. (2017) $\hat{\tau} = \frac{\tau_{i-1} + \tau_i}{2}$. In QR-DQN Dabney et al. (2017), the authors apply Huber loss to quantile loss to smoothen it. However, the solution of quantile Huber loss is different from the true quantile value. For accurate quantile regression, we trade off the converge-ability by not apply Huber loss unlike QR-DQN. For given target $y$ and prediction $y'$ error $u$ is defined by, $u = y - y'$. We

apply quantile regression to discriminator to learn a distribution of realisticity. We modify DCGAN Radford et al. (2016) architecture to output vector of $N$ dimension instead of scalar for prediction of quantile values. The target of real samples is $a$ and the one of fake samples is $b$. The output of discriminator is the realisticity distribution of the input batch. We simply define it by the mean along the batch dimension of each result as did in Wasserstein GANs. For given batch size $M$ and discriminator output $o$, $D_{\tau(batch)} = \frac{1}{M} \sum_{i=1}^{M} o_{i,\tau}$. Objective functions of discriminator can be formulated as the following:

$$\min_D V_{\text{QRGAN}}(D) = \frac{1}{N} \sum_{\hat{\tau} \in (\hat{\tau_1},..,\hat{\tau_N})} \left( \rho_{\hat{\tau}}(D_{\hat{\tau}}(x_{real}) - a) + \rho_{\hat{\tau}}(D_{\hat{\tau}}(x_{fake}) - b) \right) \quad (4)$$

Intuitively, "realisticity" should not be bounded to the specific number. In that sense, we find that it works best to use $a = +\infty$, $b = -\infty$ with regularization (explained below). The quantile values will have gradients of $\tau$ for positive changes and gradients of $(\tau - 1)$ for negative changes. Still, the quantile values can be viewed as pessimistic/optimistic neurons that are mentioned in Dabney et al. (2020). The discriminator output will grow forever and thus the discriminator will not converge when we use targets $a = +\infty$ and $b = -\infty$ without regularization. We add $L1 - square$ penalty term of the discriminator outputs for convergence,

$$\min_D V_{\text{QRGAN}}(D) = \frac{1}{N} \sum_{\hat{\tau} \in (\hat{\tau_1},..,\hat{\tau_N})} \left( \rho_{\hat{\tau}}(D_{\hat{\tau}}(x_{real}) - a) + \rho_{\hat{\tau}}(D_{\hat{\tau}}(x_{fake}) - b) \right) +$$
$$\lambda \frac{1}{N} \sum_{\hat{\tau} \in (\hat{\tau_1},..,\hat{\tau_N})} \left( \frac{1}{2M} \left( \sum_{i=1}^{M} |o_{i,\tau,real}| + \sum_{i=1}^{M} |o_{i,\tau,fake}| \right) - k \right)^2 \quad (5)$$

Where, $\lambda$ and $k$ are hyperparameters. When added $L2 - penalty$ term, the discriminator outputs eventually saturate (gradients at $0$), and the generator training gets slowed. For generator to minimize $1 - Wasserstein$ distance,

$$W_1(U, Y) = \int_0^1 |F_Y^{-1}(\omega) - F_U^{-1}(\omega)|^1 d\omega \quad (6)$$

we compute the quantile values $D_{\hat{\tau}}(x_{real})$ and $D_{\hat{\tau}}(x_{fake})$ and minimize the differences using mean absolute error loss.

$$\min_G V_{QRGAN}(G) = \frac{1}{N} \sum_{\hat{\tau} \in (\hat{\tau_1},..,\hat{\tau_N})} |D_{\hat{\tau}}(x_{real}) - D_{\hat{\tau}}(x_{fake})| \quad (7)$$

We replace $D_{\hat{\tau}}(x_{real})$ by $\infty$ to prevent it updating to decrease the discriminator output when some quantile values of fake batch are often greater than that of real batch.

$$\min_G V_{QRGAN}(G) = \frac{1}{N} \sum_{\hat{\tau} \in (\hat{\tau_1},..,\hat{\tau_N})} |\infty - D_{\hat{\tau}}(x_{fake})| \quad (8)$$

Minimizing this objective function is same to maximize the following function, thus:

$$\max_G V_{QRGAN}(G) = \frac{1}{N} \sum_{\hat{\tau} \in (\hat{\tau_1},..,\hat{\tau_N})} D_{\hat{\tau}}(x_{fake}) \quad (9)$$

The proposed QRGAN is developed as shown in Algorithm 1.

### 2.3 COMPARISON TO OTHER GANS

Derivative of discriminator loss when $a = \infty$, $b = -\infty$, $N = 1$, is $0.5$ which is proportional to the derivative of the half of the critic loss of Wasserstein GANs. The two even regularizes the discriminator/critic outputs to prevent endless growth. In addition, the derivative of generator loss is $1$, same as Wasserstein GANs. This fact may mean QRGANs are an upgraded version of Wasserstein GANs that do not need to meet the Lipschitz condition for approximating 1-Wasserstein distance thanks to quantile regression.

---

**Algorithm 1: *QRGAN***

**Input:** $\eta, M, N, \lambda, k, \hat{\tau_1}, .., \hat{\tau_N}, \theta_D, \theta_G$

$\triangleright$ $\eta$ (learning rate), $M$ (batch size), $N$ (number of quantile values), $\lambda$ (regularization coefficient), $k$ (norm target)

$\triangleright$ $\hat{\tau_1}, .., \hat{\tau_N}$ (quantile values), $\theta_D$ (initial discriminator weights), $\theta_G$ (initial generator weights)

---

1: **while** *(Parameters **NOT** converged)* **do**

2:     Sample $\{x_i\}_{i=1}^{M} \sim \mathbb{P}_r$ $\triangleright$ A batch of real data

3:     Sample $\{z_i\}_{i=1}^{M} \sim p(z)$ $\triangleright$ A batch of the prior distribution

4:     $g_D \leftarrow \nabla_{\theta_D}$

$$\left[ \frac{1}{M.N} \sum_{i=1}^{M} \sum_{\hat{\tau} \in (\hat{\tau_1},..,\hat{\tau_N})} \left( \rho_{\hat{\tau}} \left( D_{\hat{\tau}}(x_i; \theta_D) - \infty \right) + \rho_{\hat{\tau}} \left( D_{\hat{\tau}}(G(z_i; \theta_G); \theta_D) + \infty \right) \right) + \right.$$

$$\left. \lambda \frac{1}{N} \sum_{\hat{\tau} \in (\hat{\tau_1},..,\hat{\tau_N})} \left( \frac{1}{2.M} \left( \sum_{i=1}^{M} |o_{i,\tau,real} + \sum_{i=1}^{M} |o_{i,\tau,fake}| \right) - k \right)^2 \right]$$

5:     $g_G \leftarrow \nabla_{\theta_G} \left[ \frac{1}{M.N} \sum_{i=1}^{M} \sum_{\hat{\tau} \in (\hat{\tau_1},..,\hat{\tau_N})} D_{\hat{\tau}}(G(z_i; \theta_G); \theta_D) \right]$

6:     $\theta_G \leftarrow \theta_D - \eta.\text{Adam}(\theta_D, g_D)$

7:     $\theta_G \leftarrow \theta_G + \eta.\text{Adam}(\theta_G, g_G)$

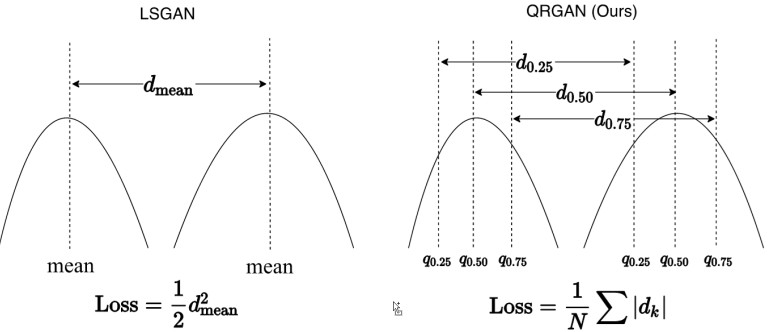

Figure 1: Comparison of LSGAN and QRGAN generator loss. LSGAN generator minimizes the difference of the means of distributions. QRGAN generator minimizes the difference of the quantile values of distributions.

## 3 EXPERIMENTS AND RESULTS

In order to investigate the QRGAN's robustness against mode collapse problem, we performed experiments using several GAN methods (NSGAN, LSGAN, QRGAN) on mixture of gaussian dataset (Ring-8, Grid-25) in the works Section 3.1. We also analyzed the discriminator output space to see if the discriminator guides the generator as expected. Also, we pinpoint the reasons why we should not use L2 penalty term in QRGANs. Furtheremore, to examize the QRGAN's generation capability, we performed different image generation experiments (CIFAR-10, LSUN-Bedroom, Cats). We evaluated the GAN methods by Frechét Inception Distance Heusel et al. (2018). We also conducted analogous analyses of QRGAN with regard to WGAN-GP using no-batch-normalization discriminator. For all image generation experiments, we only take a training set for training and evaluation. All model weights of convolutional and transposed convolutional layers are initialized from $N(0, 0.02)$ and biases of them are initialized to *zeros*. We picked Adam Kingma & Ba (2017) optimizer with $\beta_1 = 0.5$ and $\beta_2 = 0.99$ for optimization purposes. For WGAN-GP, we use $\lambda$ (gradient penalty coefficient) of 1 for CIFAR-10, and 10 for LSUN-Bedroom.

### 3.1 MIXTURE OF GAUSSIAN DATASET

To test the robustness against to mode collapse, we did the same experiments which are done in Unrolled GAN (Ring-8) and VEEGAN (Grid-25) Metz et al. (2017) Srivastava et al. (2017). We define two training dataset generator, Ring-8 and Grid-25. Ring-8 dataset contains samples from mixture of 8 gaussians arranged in ring shape. The ring has radius of 2 and each gaussian distribution has standard deviation of 0.02. Meanwhile, Grid-25 dataset contains samples from mixture of 25 gaussians arranged in grid $(5x5)$ shape. The centers of gaussian distributions range from $[-4, -4]$ to $[4, 4]$. Each distribution has standard deviation of 0.05. The MLP-based model architecture used for MoG experiments is shown in Fig. 7. Hyperparameters used for Ring-8 and Grid-25 experiments are presented in Table 2. The experiment results are shown in Fig. 2.

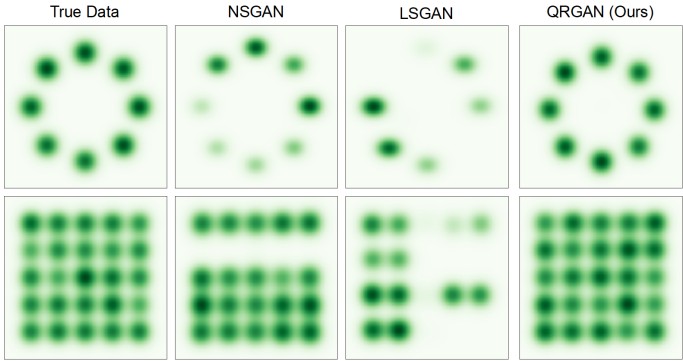

Figure 2: (Top) Ring-8 experiment result. (Bottom) Grid-25 experiment result.

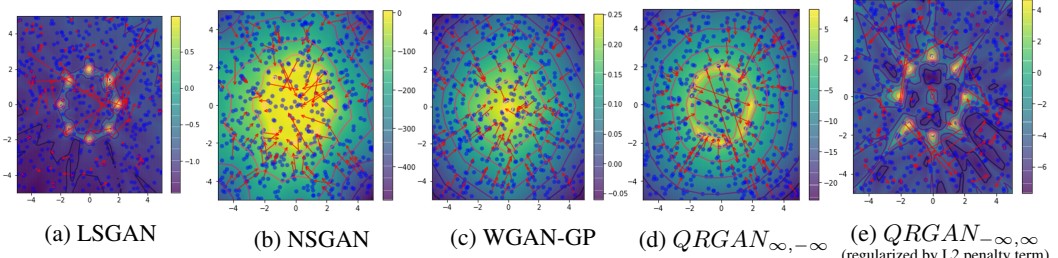

Figure 3: Contour plots of discriminator output space

Mode collapse occurred on NSGAN and LSGAN in both Ring-8 and Grid-25 experiments. However, mode collapse is not observed on our proposed method (QRGAN) experiments. Mode collapse occurs by various reasons. One of the most decisive reason is local minima in the space created by discriminator. To analyze the discriminator output space at a situation which discriminator overwhelms generator, we trained discriminator by various GAN methods with dummy generator which outputs samples from uniform distribution. The color is the discriminator output value (higher is more realistic). Blue dots are dummy generator output, sampled from uniform distribution. We pick some blue dots and draw red arrows (force) by normalized computed gradients by generator loss. The contour plots of discriminator ourput spaces are shown in Fig. 3. As we can see from the contours, very steep slope appears only near real samples and very gentle slope appear for samples outside. Thus, gradient norm is very small (the arrow length is short). Most gradient directions are not desirable for fake samples. Looking at the colors, discriminator guides proper Wasserstein distance minimization. Slopes are not too gentle fake samples. Red arrows consistently head to real samples. Like LSGAN case, gradients are too gentle for fake samples and gradient directions are decided by noise, thus not desirable. Generator is likely to suffer from local minima. In common, discriminator is modeled to predict probability. For the optimal discriminator, there are no output difference between real samples. This behavior is same to fake samples, too. In real, noise makes difference, thus gradient directions are noisy. Then, generator is not trained to make realistic samples. This works very similarly for LSGANs and bounded (which use finite targets) QRGANs,

too. Instead, we can model a discriminator to predict distance. If discriminator predicts distance, it should be less affected by noise.

### 3.2 IMAGE GENERATION EXPERIMENTS

#### 3.2.1 EVALUATION METRICS

We use Fréchet Inception Distance Heusel et al. (2018) (FID) for evaluation of image generation experiments. FID is an evaluation metric for image generation models. Although Inception Score (IS) is used previously, there are cases where IS evaluates worse than FID because does not reflect training data. FID calculates 2048-dimension feature vector from Inception-v3 model. It calculates mean and covariance matrix from the feature vector, and then compute Fréchet distance from the statistics. Lower FID value means that the two statistics are closer. Given mean $\mu_1$ and $\mu_2$, and covariance matrix $\Sigma_1$ and $\Sigma_2$, the Frechet inception distance could be represented as following, $FID = \|\mu_1 - \mu_2\|^2 + Tr(\Sigma_1 + \Sigma_2 - 2\sqrt{\Sigma_1 \Sigma_2})$. We used the dataset and approach presented in Heusel et al. (2018) for FID value calculation. We generate 50000 images and then calculate FID value for each 4k iterations and compare the minimum Fréchet Inception Distance (FID) value.

#### 3.2.2 ARCHITECTURE FOR EXPERIMENTS

**Deep Convolutional GANs:** Deep Convolutional Generative Adversarial Networks Heusel et al. (2018) (DCGANs) introduce rules for stable adversarial training of convolutional networks. Additionally, the authors have shown vector arithmetic operations (addition, subtraction) on the latent code representations and it means that they successfully vectorized image. For image generation experiments, we use DCGAN-based architectures as shown in Fig. 8.

**Checkerboard artifacts and Resize-convolution:** In DCGAN, generator outputs with checkerboard artifacts when the kernel size is not divisible by stride Odena et al. (2016). These artifacts are generated because the outputs can be unevenly overlap. This phenomenon also occurs in backward pass of convolution operations. To fix the checkerboard artifacts, we use convolution-average-pooling and resize-convolution layers. For downsample pass in the discriminator, we use kernel size of 3, stride of 1, padding of 1 for convolutional layers and following average pool 2d layer. For upsample pass in the generator, we upsample the input by nearest neighbor method and then pass it to following convolutional layer. The checkerboard artifacts is presented in Fig. 9. The parameters of different architecture variants are shown in Table 3. Hyperparameters are presented in Table 4. The results of computed minimum FID are as in Table 1. QRGAN has shown better FID values than NSGAN and LSGAN and competitive score to WGAN-GP. By applying the regularization method learned from the discriminator output space analysis, gradients keep good norm and direction. Although WGAN-GP shows great performance in several experiments, the computation is very costly and take much more iterations. The analysis results for comparison of QRGAN against NSGAN, LSGAN and WGAN-GP are shown in Fig. 4, Fig. 5 and Fig. 6 for the experiments on CIFAR-10, LSUN-Bedroom and Cats, respectively.

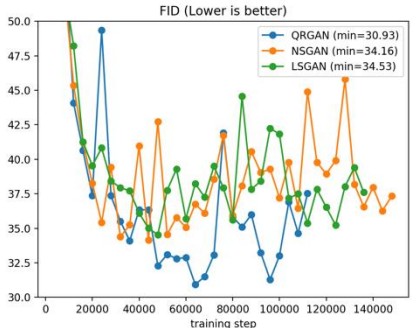

(a) QRGAN compared to NSGAN and LSGAN

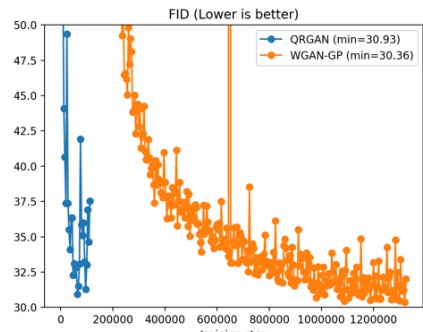

(b) QRGAN compared to WGAN-GP

Figure 4: CIFAR-10 generation experiment results

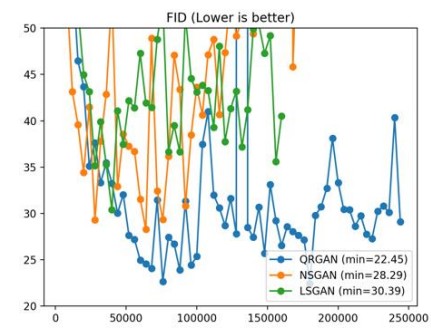

(a) QRGAN compared to NSGAN and LSGAN

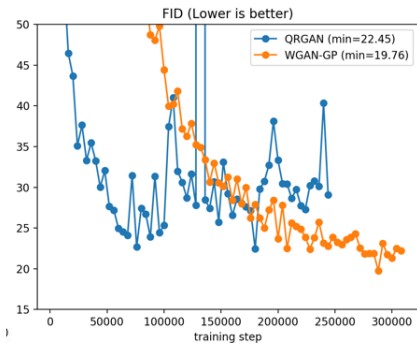

(b) QRGAN compared to WGAN-GP

Figure 5: LSUN-Bedroom generation experiment results

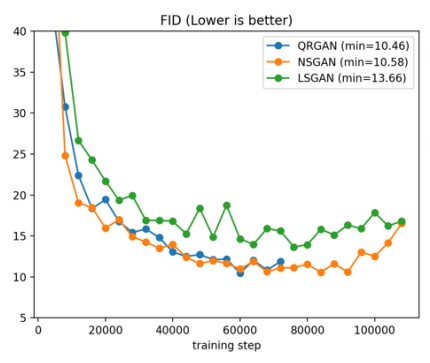

(a) QRGAN compared to NSGAN and LSGAN

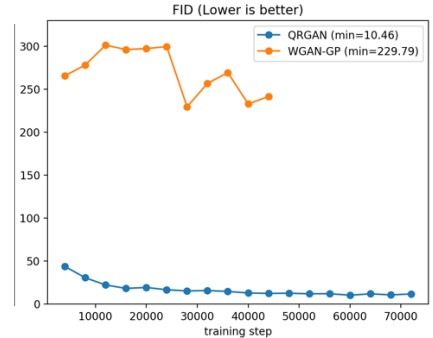

(b) QRGAN compared to WGAN-GP

Figure 6: Cats generation experiment results

Table 1: The minimum FID summary

| Minimum FID | CIFAR-10 | LSUN-Bedroom | Cats |
|-------------|----------|--------------|--------|
| NSGAN       | 34.16    | 28.29        | 11.37  |
| LSGAN       | 34.53    | 30.39        | 13.68  |
| WGAN-GP     | **30.36**| **19.76**    | 229.8  |
| **QRGAN**   | 30.93    | 22.45        | **10.46** |

## 4 REMARKS

We presented the proposed QRGAN, a new method to train GANs that minimize 1-Wasserstein distance between real data distribution and fake data distribution using quantile regression. By modeling a discriminator to predict quantiles, more details could be learnt. We explained how mode collapse is created, and what is the desired characteristics of discriminator output space. We showed that LSGAN discriminator creates sharp local minima, thus it can get stuck in mode collapse. On the contrary, WGAN-GP did not create any sharp local minima. We found an appropriate regularization method from this analysis. The result has shown improved generation quality than previous state of the art GAN methods.

ACKNOWLEDGMENTS

This research was partially supported by Basic Science Research Program through the National Research Foundation of Korea(NRF) funded by the Ministry of Education(No. 2020R1A6A1A03046811). This research was partially supported by the MSIT(Ministry of Science, ICT), Korea, under the ITRC(Information Technology Research Center) support program(IITP-2020-2016-0-00465) supervised by the IITP(Institute for Information & communications Technology Planning & Evaluation)

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

## APPENDIX

## A   FIGURES AND TABLES RELATED TO EXPERIMENTS

Due to the space limitation, we present related figures and tables of our experiments in appendices.

### A.1   THE MLP-BASED MODEL ARCHITECTURE USED FOR MOG EXPERIMENTS

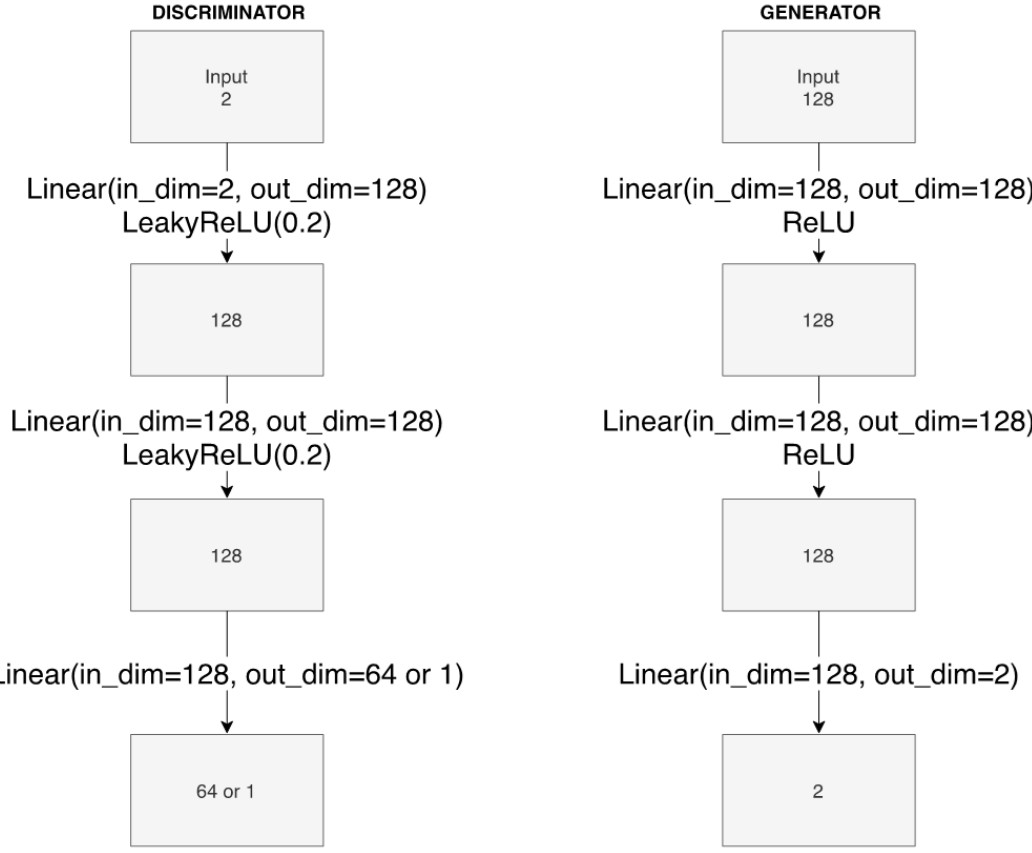

Figure 7: The MLP-based model architecture used for MoG experiments

## A.2   CIFAR-10 ARCHITECTURE

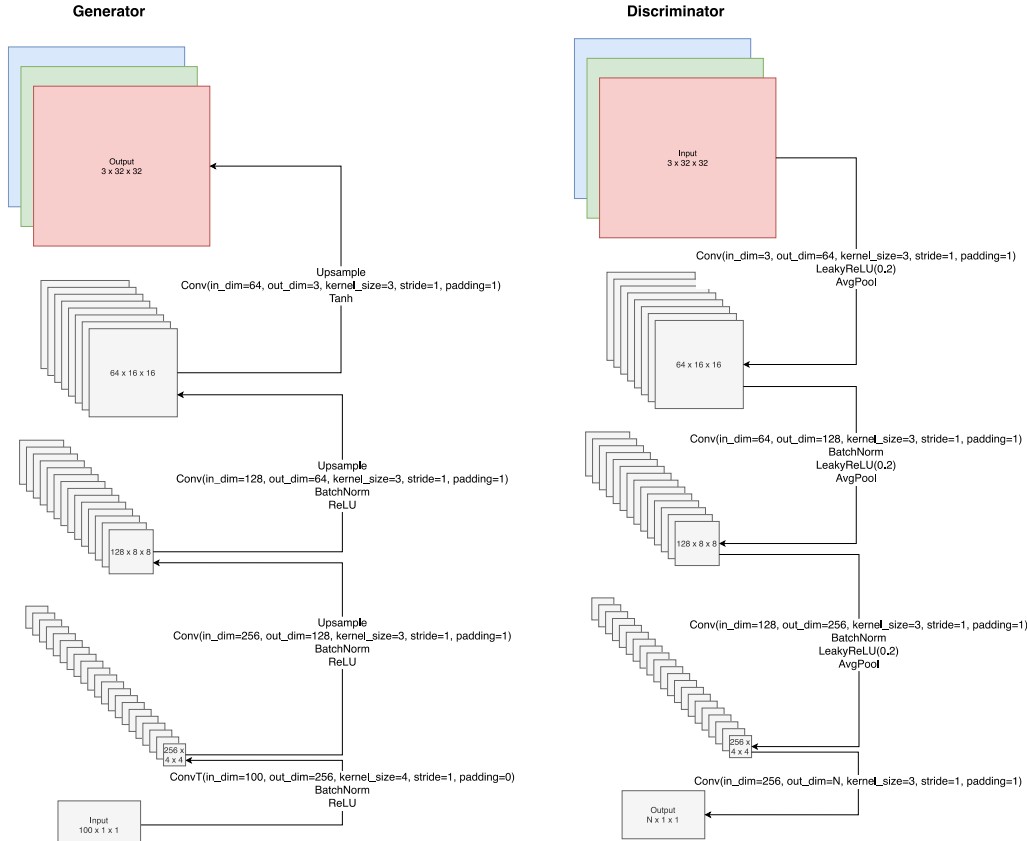

Figure 8: Architecture for CIFAR-10

### A.3 CHECKERBOARD ARTIFACTS AND RESIZE-CONVOLUTION

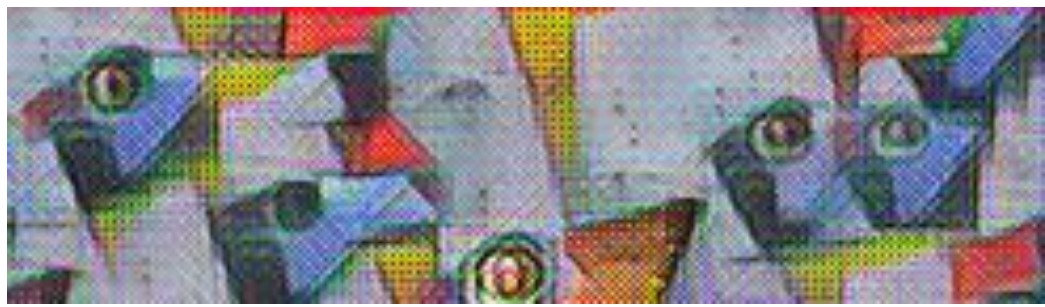

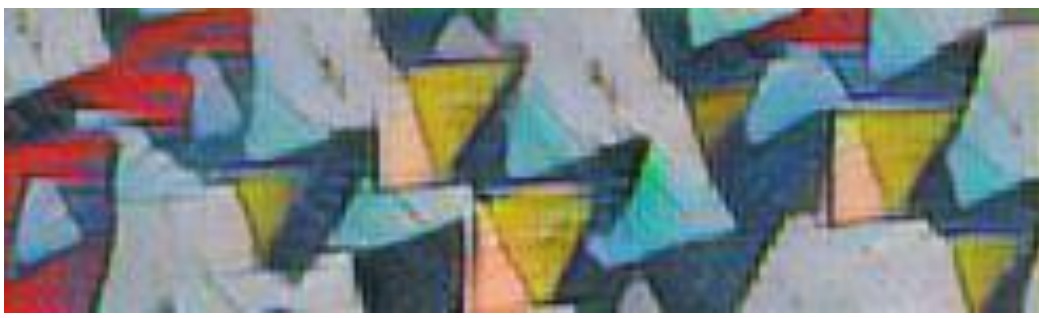

Figure 9: Checkerboard artifacts. (Top) Checkerboard artifacts has caused because kernel size is not divisible by stride. (Bottom) No checkerboard artifacts appeared by using resize-convolution.

## A.4 TABLES OF PARAMETERS

Table 2: Hyperparameters used for Ring-8 and Grid-25 experiments

| Hyperparameter Name | Value |
|---|---|
| Batch size | 500 |
| Dimension for latent vector | 128 |
| Discriminator Model | MLP without BN |
| Generator Model | MLP without BN |
| Main dimension for discriminator | 128 |
| Main dimension for generator | 128 |
| Learning rate for discriminator | 1e-3 |
| Learning rate for generator | 1e-3 |
| Optimizer | Adam |
| $\beta_1$ (Adam) | 0.5 |
| $\beta_2$ (Adam) | 0.99 |
| Number of quantiles | 64 |
| $\lambda$ (L1-square penalty) | 0.1 |
| k (Norm target) | 0 |

Table 3: Architecture variants summary

| Task | Number of hidden layers | Image size |
|---|---|---|
| CIFAR-10 | 3 | 32x32 |
| LSUN-Bedroom | 4 | 64x64 |
| Cats | 4 | 128x128 |

Table 4: Hyperparameters summary

| Hyperparameter Name | CIFAR-10 | LSUN-Bedroom | Cats |
|---|---|---|---|
| Batch size | 64 | 64 | 64 |
| Image size | 32x32 | 64x64 | 128x128 |
| Dimension for latent vector | 100 | 100 | 100 |
| Main dimension of discriminator | 64 | 64 | 64 |
| Main dimension of generator | 64 | 64 | 64 |
| Discriminator Model | DCGAN | DCGAN | DCGAN |
| Generator Model | DCGAN | DCGAN | DCGAN |
| Learning rate for discriminator | 2e-4 | 2e-4 | 5e-5 |
| Learning rate for generator | 2e-4 | 2e-4 | 5e-5 |
| Optimizer | Adam | Adam | Adam |
| $\beta_1$ | 0.5 | 0.5 | 0.5 |
| $\beta_2$ | 0.99 | 0.99 | 0.99 |
| Number of quantiles | 64 | 64 | 64 |
| $\lambda$ (L1-square penalty) | 0.1 | 1.0 | 0.1 |
| K (Norm target) | 0.0 | 1.0 | 0.0 |

## B  BACKGROUND AND PRELIMINARY

We present the background and preliminary for this study as in the following appendices for clearer understanding of the proposed QRGAN.

### B.1  DISCRIMINATION AND GENERATION

Generally, generation tasks are known to be complex than classification/regression tasks because the model learns joint probability distribution: Given an observable variable $X$ and a target variable $Y$:

$$p(X, Y) = p(X|Y).p(Y) \tag{10}$$

while discriminative model learns only $p(X|Y)$.

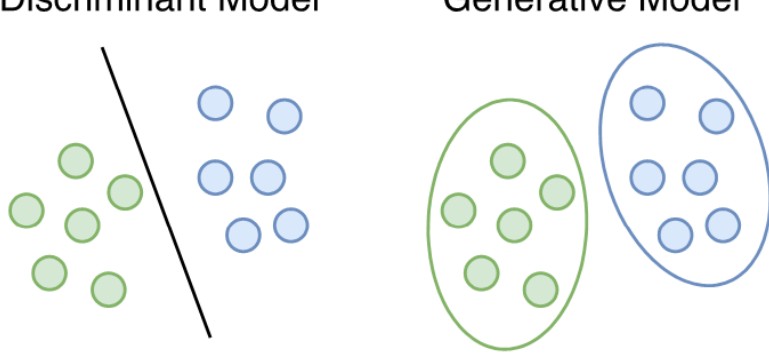

Figure 10: Discriminant model vs generative model

There are many attempts to train neural-network-based generative models like Variational Autoencoders (VAE), Generative Adversarial Networks (GANs), and Adversarial Autoencoders. Kingma & Welling (2014) Kingma & Welling (2014) Makhzani et al. (2016)

### B.2  THE KULLBACK-LEIBLER (KL) DIVERGENCE

The Kullback-Leibler (KL) divergence is defined as follows,

$$D_{KL}(P|Q) = \int p(x)log\left(\frac{p(x)}{q(x)}\right)dx \tag{11}$$

where both $P$ and $Q$ are assumed to be continuous, is an integral probability metric (IPM) to measure diversity between $P$ and $Q$. The KL divergence can be infinite when there are points where $P(x) = 0$ and $Q(x) > 0$. In that case, gradients calculated by KL divergence tends to be zero, then training gets slow.

For GAN discriminator loss,

$$\bigtriangledown \sigma(x) = 0 \text{ where } \sigma(x) = 0 \text{ or } \sigma(x) = 1 \tag{12}$$

For NSGANs, Non-saturating GANs, whose generator loss is defined as:

$$\min_{G} V_{NSGAN}(G) = -log\sigma\left(D(x_{fake})\right) \tag{13}$$

Then, $log0 = -\infty$, where $\sigma(x) = 0$

### B.3 WASSERSTEIN DISTANCE

To fix the non-convergence problem of the KL divergence, there are attempts to minimize Wasserstein distance in recent work. The $p - Wasserstein$ metric $W_p$, for $p \in [1, \infty]$, also known as the Earth Mover distance when $p = 1$ Levina & Bickel (2001), is an integral probability metric (IPM) between distributions. The $p - Wasserstein$ distance is $L^p$ metric on inverse cumulative distribution functions between distributions $U$ and $Y$.

$$W_p(U, Y) = \left( \int_0^1 |F_Y^{-1}(\tau) - F_U^{-1}(\tau)|^p d\tau \right)^{\frac{1}{p}} \tag{14}$$

where for a random variable $Y$, the inverse CDF $F_Y^{-1}$ of $Y$ is defined by,

$$F_Y^{-1}(\omega) := inf \, y \in R : \omega \le F_Y(y) \tag{15}$$

where, $F_Y(y) = Pr(Y \le y)$ is the CDF of $Y$ Dabney et al. (2017). Wasserstein distance is difficult to compute because of its complexity. In Wasserstein GANs, the authors apply Lipschitz condition which is defined by,

$$d_A \left( f(x_1), f(x_2) \right) \le K d_B(x_1, x_2) \tag{16}$$

for given metrics $d_A$ and $d_B$ to apply an approximation method for the 1-Wasserstein distance. This constraint limits the neural network capacity because the model architecture and weight value should be restricted.

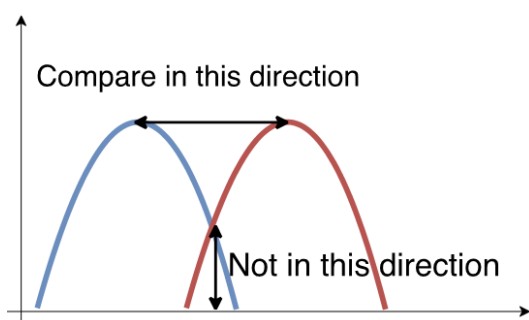

Figure 11: 1-Wasserstein distance is also called Earth Mover's distance

### B.4 VARIATIONAL AUTOENCODER

#### B.4.1 AUTOENCODER

Autoencoder is a neural network architecture used to learn latent representations in an unsupervised manner. It consists of encoder and decoder networks, and usually encoder's output dimension is smaller than input dimension, called bottleneck.

Objective function of autoencoders can be formulated as follows:

$$\min_\theta J_{AE}(\theta) = \frac{1}{2} \left( X - D(E(X; \theta); \theta) \right)^2 \tag{17}$$

where $E$ is encoder network outputs code and $D$ is decoder network which predicts input to the encoder. To generate samples from the decoder, we should generate appropriate input for the decoder. However, it is very difficult to generate the input due to the intractability of the latent distribution of plain autoencoders.

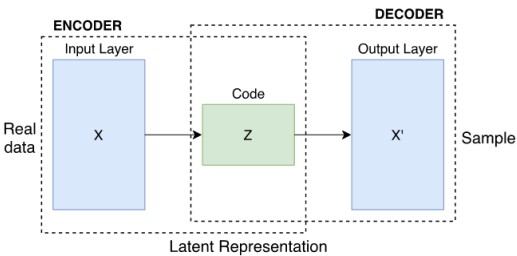

Figure 12: Autoencoder architecture

### B.4.2 VARIATIONAL AUTOENCODER

In VAEs, they make the encoder to output parameters of a tractable distribution such as normal distribution. By doing so, we can calculate Kullback-Leibler (KL) Divergence of the code distribution and a prior distribution. For given encoder $E$, decoder $D$, prior distribution $p$, the objective is:

$$\min_\theta J_{VAE}(\theta) = \frac{1}{2}\left(X - D(Z;\theta)\right)^2 + KL(Z\|p) \tag{18}$$

where, $\mu, \sigma = E(X;\theta)$ and $Z \sim N(\mu, \sigma I)$.

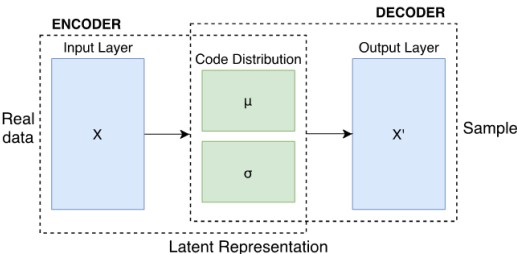

Figure 13: Variational Autoencoder architecture

In addition, reparameterization trick is applied for backpropagation. $Z = \mu + \sigma.\epsilon$, where $\epsilon \sim p$. In VAEs, the KL divergence regularization assumes every code distribution to be $p$ even when different input is given. This leads to input aliasing to the decoder and worsen blurry outputs.

### B.5 GENERATIVE ADVERSARIAL NETWORKS

### B.5.1 GAN

Recently, Generative Adversarial Networks (GANs) have been introduced. Discriminator learns whether the input is real or fake, and generator learns to fool discriminator to generate realistic samples by help of discriminator. GANs successfully solved problems of VAEs. Unlike VAEs, GANs can hypothesize the entire code distribution. Adversarial autoencoders (AAEs) take this idea to make generative autoencoder which regularizes the entire code distribution. GANs can be formulated as the following:

$$\min_G \max_D V(D, G) = \mathbb{E}_{x \sim p_{dt}(x)}\big[logD(x)\big] + E_{z \sim p_z(z)}\big[log(1 - D(G(z)))\big] \tag{19}$$

Because gradients could be very small at the beginning, alternative loss function for generator, namely NSGAN is proposed.

$$\max_G V(G) = \mathbb{E}\big[logD(G(z))\big] \tag{20}$$

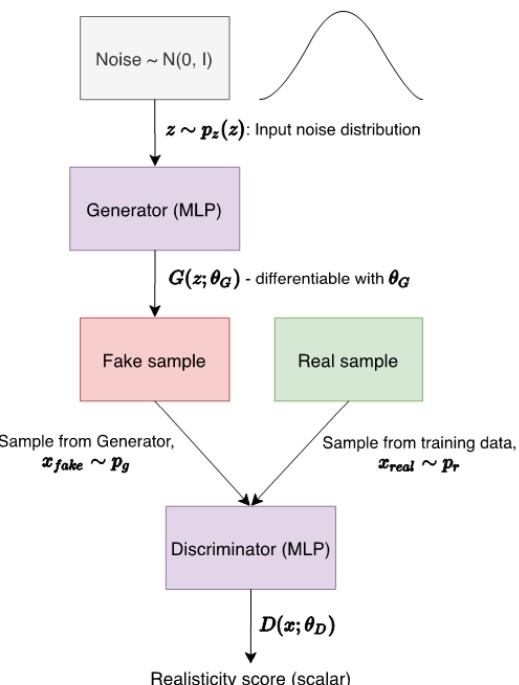

Figure 14: The Generative Adversarial Networks framework architecture

### B.5.2 DIFFICULTY OF TRAINING GANS

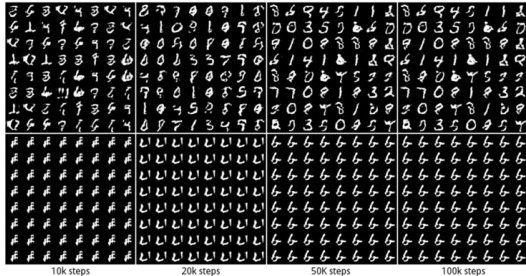

Figure 15: (Upper, Unrolled GAN) the GAN training is successful without mode collapse. (Lower, Standard GAN) the GAN training is failed, and this type of GAN failure is called mode collapse that generator outputs very similar images Metz et al. (2017)

GANs are difficult to train because the networks compete each other. Because discriminator is used for generator loss calculation, the loss can oscillate and be unstable. In addition, cross entropy loss used in the original GANs often saturates. When failed to train GANs, mode collapse problem often occurs.

### B.6 REINFORCEMENT LEARNING

Reinforcement Learning is to find a policy that maximizes given rewards by environment. Given state and action, an environment outputs next state and reward. Agent should try various action

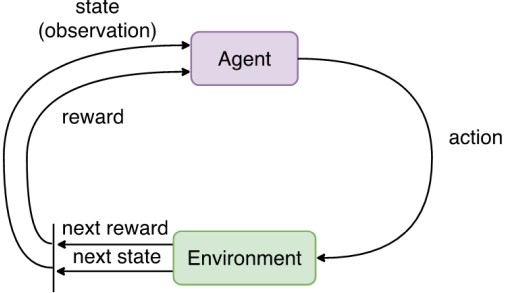

Figure 16: Reinforcement learning

sequence to find best policy. $Q - Learning$ is learning Q-values iteratively. $Q - value$ is defined as,

$$Q(s, a) = \mathbb{E}\big[\sum_{t=0}^{\infty} \gamma^t R_t | s, a\big] \tag{21}$$

and this can be represented by Bellman equation,

$$Q(s, a) = R + \mathbb{E}\big[\gamma Q(s', a')\big] \tag{22}$$

where $s$ is current state, $a$ is the action, $r$ is reward, $a'$ is the next action, and $\gamma$ is a hyperparameter called discount factor to discount future rewards exponentially.

### B.6.1 DEEP Q NETWORK (DQN)

Deep Q Network Mnih et al. (2015) successfully applies deep learning to solve the Bellman equation. The training is unstable because the Bellman equation is recursive. Target network, a cloned network only for calculation of target Q value, is used to stabilize the training. Moreover, replay memory is applied to randomly sample the training data.

### B.6.2 C51

Moreover, there was an attempt to learn a value distribution instead of an expected value of it. In C51, the authors define three hyperparameters: the number of supports, low and high for Q-value. By making bins by the three hyperparameters, the regression task has been turned to a classification task. Unlike DQN, target is not a scalar value but a distribution. To optimize predicted distribution to target distribution, they projected supports properly to fit bins then minimized the KL divergence between the two. Learning value distribution stabilizes training and improved prediction performance. Although they prove that 1-Wasserstein distance should be minimized because KL divergence is not mathematically guaranteed to converge, they could not find a way to minimize it.

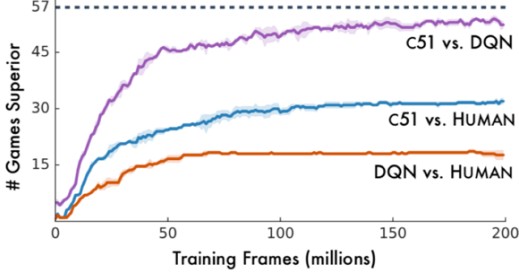

Figure 17: C51's performance compared to human and DQN Bellemare et al. (2017)

### B.6.3 Quantile Regression Deep Q Network (QR-DQN)

QR-DQN Dabney et al. (2017) is proposed to fix the non-convergence problem of C51. The authors hypothesize the probability distribution function (PDF) consists of N dirac delta functions. We can optimize the PDF by changing the supports of the dirac delta functions. Likewise, the CDF can be also optimized by quantiles.

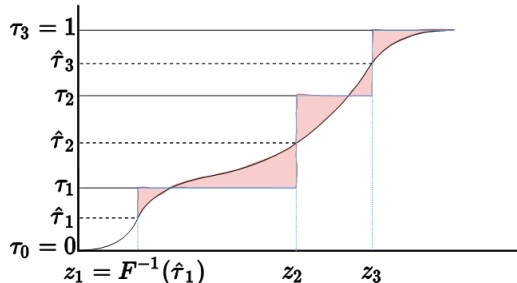

Figure 18: The CDF which is represented by 3 Dirac delta functions. Area filled by red represents the 1-Wasserstein distance. The 1-Wasserstein distance is proportional to the sum of difference between quantile values at each quantile fractions. Therefore, quantile regression is the equivalent to 1-Wasserstein distance minimization.

