# OpenReview forum: "QRGAN: Quantile Regression Generative Adversarial Networks"
_ICLR.cc/2021/Conference — Reject_

### Official Review · AnonReviewer4 · 2020-10-25
**Review for QRGAN**

**Rating:** 2
**Confidence:** 5

**Review:**

1. This paper contains many typos, grammar mistakes, and format problems, which make it very hard to read. For most equations, there is no ending period. The citation format seems wrong.

2. The equations (1) and (2) are for 1-dimensional random variables. They may not be suitable for high dimensional random variables. The $N$ quantile values defined in this paper are for which random variable?

3. For the neural network for $D_\tau$, does it mean the input is $(\tau, X)$? So for different $\tau$, they share the same weights. What is the meaning of this output?

4. The "$+\infty$" and "$-\infty$" notation is confusing. Does this mean we choose a very big value or a very small value in practice? But this is very subjective now. Did you perform some sensitive analysis on the choices of $a$ and $b$?

5. The authors claim that the WGAN training is slow. For WGAN-GP, I don't see why the training is much slower than the training of QRGAN. Did you perform some analysis on the training time?

6. The arguments for QRGAN  to overcome mode collapse is quite vague. Many GANs with the encoder structure can solve the mode collapse well. It may benefit to compare  QRGAN with these methods.

7. What is the implication of Table 1? Does that mean WGAN-GP is better than QRGAN? The generative images are not demonstrated. Other dataset such as CelebA can be applied to check the performance. Image interpolation can also be demonstrated for mode collapse situation.

8. The Appendix B is completely not necessary. It contains only well known results.

---

### Official Review · AnonReviewer3 · 2020-10-28
**Promising Idea, but experiments need more work**

**Rating:** 4
**Confidence:** 4

**Review:**

Summary. The paper proposes to use quantile regression as an alternative GAN loss. The idea is to evaluate the quantiles of discriminator outputs instead of just using a single scalar discriminator value: while all quantile discriminator outputs for real/fake data is pushed to +infty/-infty, the generator's cost function tries to push the quantiles of fake data to be as realistic as possible (+infty). As the regression loss leads to runaway values, a L2 regularizer is put on the discrimantor output quantiles. Experimental results show very good mode coverage on Gaussian Mixtures. Further experiments demonstrate it's in the ballpark of WGAN-GP on CIFAR-10, best on CATs, and second on LSUN-Bedroom.

Reasons for score: While the paper presents a reasonable and (to me) novel idea in GANs, the experiments fall short in comparing to the state of the art, and also the ingredients in the method are not sufficiently dissected to attain a sufficient understanding. Hence, I suggest to reject the paper in its current form.

Pro.
- the paper proposes a simple and novel target function for GAN training, that makes intuitively sense
- the mode coverage on the toy Mixture of Gaussian experiments look very solid


Con.
- experiments:
   - on CIFAR-10 etc. lack comparison to state-of-the art (sota) methods, which is necesary to put the work in context. E.g. it should be added SN-GAN (Miyato et al.), StyleGAN(2).
   - similar for the Mixture of Gaussian: comparison to other standard methods in the literature that tackled this problem are missing (e.g. Unrolled GANs).
   - also, please check again WGAN-GP on cats - why is WGAN-GP seemingly not training at all (and starting at a much higher level from the start in Figure 6)?
- the regularizer in equation 5 pushes both discriminator for real and fake towards the same values - hence potentially counteracting stability issues in training. This should be investigated independently to understand the effect of this regularizer in isolation (can even be formulated also for standard GAN KL losses, e.g. by pushing the average to 0.5). Otherwise it remains unclear if the benefits of the methods are attributable to the quantile regression or this regularizer
- the exposition should be improved:
  - pg 2: too much stuff on RL - this is not needed in the paper and should be shortened to a minimum
  - the method could be written to be understandable more easily (e.g. give a high-level description of the intuition similar to my Summary above before diving deep into the formulae)


Rebuttal:
- please address the points on the Con side.


Minor issues:
- the relation to standard divergence minimization remains unclear; i.e. in particular it remains theortically unclear if this really converges to the target distribution (the empirical results seem encouraging though)
- many typos and language needs improvement - please check the document carefully again

---

### Official Review · AnonReviewer1 · 2020-10-29
**Having the discriminator output a whole distribution. Nice idea, average paper**

**Rating:** 5
**Confidence:** 4

**Review:**

# General statements
the core contribution of this paper is to train GAN by designing the discriminator so that it outputs a whole distribution instead of a point estimate for "realism". This distribution is instantiated through its quantiles, and the whole approach is thus framed in a quantile-regression framework. The nice feature is that using losses over quantiles means using a wasserstein distance, which has strong properties in a GAN setting.

The idea is interesting and is definitely worth investigating. I am not 100% sure it was not presented previously. But I trust the authors on this.
* All in all, the paper is average in terms of english usage: totally acceptable in the beginning, there is a strong degradation for the experimental section, that has been either written in haste, or by another author than the rest.
* The actual performance is rather disappointing, since the authors do not clearly manage to demonstrate any superiority of their proposed approach vs author (classical) methods, except on toy data. I don't see this as a real issue though, because the fact that the paper is inspiring is what I believe is most important.
* The quantile regression loss is not presented in a way that allows people not knowing it already to understand the paper. This must be changed.
* you didn't study the impact of the number of quantiles, although it looks like something you definitely should have done, since it's the core contribution of the paper.


# Detailed comments
Below are remarks and typos found along the way:

## Abstract
* "And we found that he discriminator should not be bounded to specific numbers." is unclear here.

## Introduction
 * "Text or structured data [...] of the world " awkward
 * "p(z|x), not p(z)" awkward
 * "Mode collapse is caused by unstable training and improper loss function" : reference ?
 * "propose to use mean square error (MSE) which does not saturate": at this stage, you didn't introduce this concept of "saturation". And you don't tell where the MSE is applied
 * "results better models" : results in better models ?
 * "Reinforcement Learning (RL):is to learn": awkward. And we don't understand why you're mentioning RL at this stage. Above, you only introduced VAE and GAN for your purpose. This paragraph comprises no reference.
 * "and gradually reach to the optimal policy.": typo
 * The reason why you introduce RL becomes clearer after you introduced QR-DQN. I think that it's however a bit weird as it is framed currently. You should mention that he inspiration and motivation of your work originates from RL and some of its recent developments.
 * "Discriminators whosetarget is specified": what do you mean ?
 * "mixture of gussian"

## Quantile Regression GAN
 * one really needs to know the trick already to understand equation (3). You must provide a reference here and explain the relationship between quantile regression and (3) as a loss.
 * "D_{\tau(batch)}" instead of "D_\tau(batch)" ? above equation (4)
* it reads rather uncommon to me to write that infinity (negative or positive) is the objective of the discriminator, with a regularization that constraints its magnitude. I suspect the reason for this to work is: you don't actually provoke some strong collapse of the discriminator output to a specific value (a or b), but rather enforce that it stays somewhere in the approximative range [-k/2M k/2M]. This looks like a nice trick. But I would have appreciated some discussion about it.
* Is there a reason why you picked 1-wasserstein rather than 2- or p-wasserstein ?
* We replace Dτ(xreal) by ∞ to prevent it updating to decrease the discriminator output": awkward sentence.
* Actually, I don't really understand (7). What is "x_real" in the setting of trainng the generator ? You just have fake samples at this stage, and you're indeed using your discriminator for computing your los. I would have written min |a-D(x_fake)|.
* maybe I'm missing something, but in Alg. 1, I don't clearly understand the difference between your notation o_{i,\tau} and D_\thau(x_i). Aren't them the same ? I understand that in your definition of D, you average over the batch. but here in this algorithm box you use a notation D(x_i), which makes it identical with o_i as far as I understand.

## Experiments and results

### toy

* "arranged in grid (5x5)": the `x` doesn't render well.
* "by normalized computed gradients by generator loss": awkward
* " ourput spaces"
* "steep slope appears": slopes appear. "very gentle slope appear": gentle slopes appear ? "gentle" reads awkaward to me.
* This whole paragraph is extremely badly written and must be written completely, from "As we can see" to "less affected by noise". The english there is very bad, I don't understand what happened out of a sudden.
* I don't understand what is depicted for (d) and (e) in figure 3: since the output of your discriminator is a whole distribution, what is it exactly that you decided to plot ? Did you pick a specific quantile ?
* "Instead, we can model a discriminator to predict distance. If discriminator predicts distance, itshould be less affected by noise." what should I understand here ? I am sorry but I really don't understand the discussion here. are you eventually discarding your model and changing it for something else that would predict a "distance", whatever it means ?

### image

* "the checkerboard artifacts is"
* how many quantiles are you using ?
* Inspecting your results on figures 4-6, I'd say they don't look particularly favorable. i/ For CIFAR10, they look kind of similar with NSGAN and LSGAN, and eventually WGAN-GP gets better. ii/ same result for LSUN, although the proposed method looks better than NSGAN and LSGAN at the end, after much instability. still catched up by WGAN-GP eventually. iii/  for cats, your method looks totally similar to NSGAN and LSGAN, and there was apparently some problem in the finetuning of WGAN-GP, that just didn't train for some reason you should have investigated. It really doesn't look like what happend for it with the other datasets. table 1 hence should not be taken too seriously, unless you really can tell that this WGAN-GP could not be made better on this "cats" dataset.

## Acknowledgments: should that be part of a double blind review ?

## References:
* are not consistent. Sometimes full names, sometimes just initials. please make consistent.

## Apendixes
I am not sure appendix B is necessary

---

### Official Review · AnonReviewer2 · 2020-10-29
**Although the proposed method is interesting, there are many errors and shortcomings in the paper.**

**Rating:** 3
**Confidence:** 4

**Review:**

The authors propose the Quantile Regression GAN (QRGAN) to minimize the 1-Wasserstein distance between the real and generated data distributions. The proposed method avoids the mode collapse problem and obtains an improvement in the FID score compared to some existing GANs.

-  Pros:
  - Compared with NSGAN and LSGAN, the proposed method avoids mode collapse and achieves better performance than them in the FID score.
  - In addition, compared to WGAN-GP, it achieves better FID with less training iterations while maintaining comparable performance.
- Cons:
  - Overall, there are so many typos and grammatical errors in this paper that they make it difficult to understand the content of the paper. The authors must look for these errors and correct them.
  - Also, the way of citing figures and tables is inappropriate. For example, Fig. 1 is not cited in the main text, and figures and tables in appendixes such as Fig. 7 and Table 2 are cited without specifying that they are in appendixes. Reviewers don't need to read the appendixes, so the content should be complete in the main text.
  - The authors state that the relationship between quantile regression and 1-Wasserstein distance is shown in section 2.1, but this is not explicitly shown. In particular, the authors state in section 2.1 that "Here, minimizing 1-Wasserstein distance is same to minimizing distance between quantile values", but Eq.1 and Eq.2 are simply p-Wasserstein distance and 1-Wasserstein distance, so it is unclear which equation represents the relationship. Also, the period in Eq. 3 should be a dot (multiplication).
  - In Eq. (4), you state that a and b are set to +∞ and -∞ respectively, but how were these infinities implemented in practice?
  - Why is there no WGAN-GP result in Figure 2? My understanding is that QRGAN minimizes 1-Wasserstein like WGAN-GP, so the result is almost the same. And why didn't the authors include unrolled GAN and VEEGAN results for comparison, even though they performed the same experiments as these papers?
  - If the authors claim that WGAN-GP is computationally expensive, they should show how much less expensive it is in QRGAN. QRGAN also requires the sum of multiple quantile values, so the more of them, the longer it should take to compute them. Also, as far as I read, there is no indication in the paper of how the number of quantile values was set up in the experiment.
  - Looking at Figure 4 and Figure 5, GRGAN appears to be less stable than WGAN-GP. Why is this?
  - In section 3.2, the authors should show the image actually generated by GANs.

- Minor comments:
  - It is difficult to read because the author's citation is not enclosed in parentheses.
  - Some parts of the random variables are in bold type and others are not. These notations should be consistent.

---

### Official Review · AnonReviewer5 · 2020-11-05
**The quantile regression to GANs looks new, but not well demonstrated by either theoretical or empirical evidence**

**Rating:** 2
**Confidence:** 4

**Review:**

– Summary –

The paper proposes a new GAN method that applies the quantile regression of reinforcement learning into GAN and aims to show this helps to estimate the 1-Wasserstein distance better without gradient regularization. The idea of quantile regression presented in the paper is a way to match two distributions like WGAN-GP yet at a more grained level and need no regularization like WGAN-GP. The experiments are conducted on 2D-toy examples (Ring-8, Grid-25) as qualitative results and three other image datasets (CIFAR-10, LSUN-Bedroom, Cats) with FID scores. The proposed method is compared with some GANs baselines: SNGAN, LSGAN, and WGAN-GP.


– Strength –

S1 - The paper proposes a new idea to apply quantile regression into GANs.


– Weakness –

W1 - The paper is not well-written, and the paper representation is not good.

W2 - The performance of the  proposed method does not look outperforming the WGAN-GP even though the paper strongly claims the robustness of this method. As shown in Fig. 4, 5, the proposed method converges faster, but is not necessarily better than WGAN-GP at the end. It looks WGAN-GP converges much more stable than the proposed method.

W3 - It does not make sense why WGAN-GP is so bad on Cats dataset as shown in Fig. 6. It could be just the problem of parameters-tuning?

W4 - It's unclear why Fig. 2 misses the WGAN-GP?

W5 – The paper does not convince me why the proposed method is better than WGAN-GP in either theoretical and empirical results. In addition, the paper does not provide sufficient theoretical content to show the 1-Wasserstein distance is the same as minimizing quantile values as claimed.

W6 - Many mathematical notions are not explained, e.g., What is $\rho_{\hat{\tau}}$ in Eq. 4? How  do the authors implement with $a = \infty$ and $b = -\infty$?. How is $o_{i, \tau}$ computed?

W7 - FID scores alone may be biased, the combination with IS is required in the experiments.

W8 – The experimental results are not sufficient, e.g., the results are with only standard DCGAN architecture, and the paper would need more ablation studies on some selected hyper-parameters, e.g., $a, b, N, k$ ... in the method.

Overall, I think the paper is far to meet the conference's standard, e.g., at paper presentation, strong empirical or theoretical evidence to justify the claims. It also would need substantial revision to improve in writing. I tend to reject the paper.

---

### Decision · Program_Chairs · 2021-01-07
**Final Decision**

**Decision:**

Reject

**Comment:**

All the reviewers agree that this paper was poorly written, which I agree upon my own reading of this paper. Section 1 is rather telegraphic and difficult to comprehend. Section 2 is cryptic in several respect, including what space of probability distributions the authors consider the Wasserstein distance, what QR task the objective function (5) for the discriminator corresponds to, especially after letting $a=+\infty$ and $b=-\infty$, and so on. The numerical experiment results do not seem convincing enough to demonstrate advantage of the proposal over existing methods. The authors did not respond to the reviews, so that many concerns raised by the reviewers have not been resolved. I would thus recommend rejection of this paper.